

# A lightweight framework for cyber risk management in Western Balkan higher education institutions

Krenar Kepuska[1] and Milo Tomasevic[2]

[1] Computing and Information Technology, Rochester Institute of Technology - Kosovo, Pristina, Kosovo
[2] Faculty of Electrical Engineering, University of Belgrade, Belgrade, Serbia

## ABSTRACT

Higher education institutions (HEIs) have a significant presence in cyberspace. Data breaches in academic institutions are becoming prevalent. Online platforms in HEIs are a new learning mode, particularly in the post-COVID era. Recent studies on information security indicate a substantial increase in cybersecurity attacks in HEIs, because of their decentralized e-learning structure and diversity of users. In Western Balkans, there is a notable absence of incident response plans in universities, colleges, and academic institutions. Moreover, e-learning management systems have been implemented without considering security. This study proposes a cybersecurity methodology called a lightweight framework with proactive controls to address these challenges. The framework aims to identify cybersecurity vulnerabilities in learning management systems in Western Balkan countries and suggest proactive controls based on a penetration test approach.

# INTRODUCTION

In the context of higher education institutions (HEIs), the learning management system (LMS) assumes a pivotal role, serving as a vital tool utilized by a diverse range of individuals, as observed (*Maryam & Mostafa, 2021*; *Josac et al., 2019*). These platforms store sensitive data, including personally identifiable information (PII) of students, email account particulars, intellectual property (I.P.), funding details, medical records, employment contracts, academic transcripts, research data, and other vital information, as underscored (*Ulven & Wangen, 2021*; *Pinheiro, 2020*). Recent investigations into information security emphasize a discernible increase in cybersecurity threats within HEIs, primarily ascribed to the decentralized structure of e-learning and the diverse composition of the user base. The primary factors behind incidents in HEIs are identified as social engineering tactics and vulnerabilities within e-learning platforms, as articulated (*Pinheiro, 2020*; *Wangen, 2019*). 35% of all data breaches in the world occur in HEIs. In addition, between 2019 and 2020, 54% of an astounding 35% of global data breaches are concentrated within HEIs, underscoring the severity of the prevailing situation. Notably, the period

Corresponding author
Krenar Kepuska, krkcad@rit.edu

spanning 2019 to 2020 witnessed a substantial 54% of HEIs in the United States reporting data breaches, as substantiated (*OpenSSF, 2022*; *Irwin, 2022*; *Chapman, 2021*). Consequently, the exigency for effective cybersecurity management has assumed paramount significance within universities and other academic establishments. This urgency is driven by the realization that the computer systems of these institutions serve as repositories for sensitive data emanating from a diverse array of users, including students, instructors, and various other personnel, as elucidated (*Pinheiro, 2020*; *OpenSSF, 2022*).

HEIs in the Western Balkan region encounter distinct cybersecurity challenges, especially in crafting and executing incident response strategies. These difficulties are influenced by the region's specific financial, technological, and educational contexts. A notable scarcity of cybersecurity expertise in the area further complicates matters. This lack of skilled professionals undermines the institutions' abilities to respond swiftly and effectively to cybersecurity incidents, leaving them vulnerable to timely threat identification, containment, and mitigation. An overarching challenge is the prevalent underestimation of cybersecurity's significance within these educational communities. The deficiency in awareness and training among faculty and students augments the risk, as the human element often becomes the weakest link in cybersecurity. Untrained individuals are more susceptible to deceptive tactics like phishing or social engineering. A pivotal aspect of cybersecurity in HEIs is protecting the confidentiality and integrity of student and staff data. The institutions often grapple with aligning their data management practices with regional and international data protection statutes, further complicating their cybersecurity landscape. To navigate these challenges effectively, a multi-dimensional strategy is essential. This strategy should bolster IT infrastructure, recruit and train cybersecurity personnel, develop versatile and robust incident response plans, and cultivate a pervasive culture of cybersecurity awareness within the academic ecosystem. Such a holistic approach is crucial for mitigating cyber risks and enhancing the overall cybersecurity posture of HEIs in the Western Balkans.

The lack of cybersecurity resilience in HEIs in Western Balkan countries can be attributed to various factors, ranging from economic constraints to infrastructural and educational challenges. HEIs in Western Balkan countries often face budgetary constraints, limiting their ability to invest in advanced cybersecurity infrastructure, tools, and technologies. There is frequently a gap in skilled cybersecurity professionals in the region. This shortage affects the institutions' capacity to manage and respond to cyber threats effectively and hampers the development of comprehensive cybersecurity strategies and training programs. Many HEIs may operate with outdated IT systems that are more susceptible to cyber-attacks. Upgrading these systems requires significant investment, which can be challenging under limited budgets. Developing and enforcing cybersecurity policies and regulations might be inconsistent or lacking. Some institutions may not have the guidance or mandate to implement robust cybersecurity measures. Political and economic instabilities in the region can impact the focus and resources allocated to cybersecurity initiatives. To address these issues and enhance cybersecurity resilience, HEIs in Western Balkan countries must focus on increasing IT infrastructure investment, improving cybersecurity education and

training, developing robust institutional policies, and fostering collaborations within the region and with international partners.

The lack of a cybersecurity incident response plan in HEIs in Western Balkan countries can be attributed to several factors that are often interrelated and stem from internal and external challenges these institutions face. One of the primary challenges is limited financial and human resources. Many HEIs in Western Balkan countries may not have the necessary funding to invest in robust cybersecurity infrastructure, including developing an incident response plan. This limitation also affects the hiring and retention of skilled cybersecurity personnel. There is often a shortage of staff adequately trained in cybersecurity practices, including incident response. This gap makes it difficult for institutions to plan for and respond to cyber incidents effectively. Cybersecurity might not be a top priority at the administrative and management levels within HEIs. This lack of awareness and understanding of cyber risks leads to less prioritization of cybersecurity measures, including incident response planning. HEIs often rely on external digital services and platforms for various functions. This interconnectivity can complicate developing an incident response plan that effectively addresses all potential points of vulnerability. HEIs may operate with outdated IT infrastructure that is more vulnerable to cyberattacks. Developing an effective incident response plan is challenging without modern and secure systems. Addressing the lack of cybersecurity incident response plans in HEIs in Western Balkan countries requires a comprehensive approach, including increased funding for cybersecurity, enhancing cybersecurity education and training, developing and enforcing relevant policies and regulations, modernizing IT infrastructure, and fostering a culture of cybersecurity awareness.

Vulnerabilities in higher education e-learning management systems (eLMSs) are another concern. As reported in *Scerbakov, Scerbakov & Kappe (2019)*, the design of these platforms is defined by the utilization of intricate hierarchical content, accommodating a wide range of users, catering to various types of materials, and a variety of programming languages. The most common learning management platforms' features include uploading and downloading of posting students' assignments, test papers, test scores, project reports, and other resources from instructors; forum discussions on various themes; databases with grading systems and actual grades; and incorporation of third-party into online learning (*Ramani, 2017*). Some of the most web application vulnerabilities are: improper input validation such as cross-site scripting, cross-site request forgery, structured query language injection, improper authentication, improper privilege management, certificate validation, and uploading of files of a potentially hazardous nature is permitted (*SANS Institute, 2021*). In addition, according to *OWASP (2021a)* and *Riadi, Umar & Sukarno (2018)*, the most prevalent web application vulnerabilities in 2021 are the following: injection flaws, broken authentication and access control, security misconfigurations, and sensitive data exposure. Furthermore, LMSs can be affected by various logical and technical vulnerabilities, for example, input validations, cross-site scripting, insecure configuration, and broken authentication (*Invicti, 2021*; *Imperva, 2021*).

Data breaches are one of the most severe concerns in HEIs, and they report increasing security incidents. Reports from the cybersecurity industry (*Cisco, 2023*; *Invicti, 2021*;

*StealthLabs, 2021*) claimed that HEIs have the highest rate of ransomware and phishing fraud compared to all other attacks in recent years. According to *McKenzie (2021)*, in 2021, critical data from a number of U.S. universities was recently revealed on the dark web. Ransomware attacks have increased by seven times in 2020 compared to 2019 in HEIs (*Liluashvili, 2021*; *IBM, 2021*). In recent years, there has been an increasing amount of literature on cybersecurity data breaches. Various studies (*Chapman, 2021*; *Bongiovanni, 2019*; *Riadi, Umar & Sukarno, 2018*; *StealthLabs, 2021*) have shown that the most prevalent flaws in HEIs include social engineering or ransomware attacks, vulnerabilities in LMS platforms, and a lack of cybersecurity standards in place.

The migration from traditional classroom instruction to virtual learning environments has precipitated a myriad of cybersecurity conundrums for scholastic entities. This evolution, hastened by the global health crisis, has augmented the breadth of educational engagements within the digital realm, consequently amplifying the susceptibility to cyber incursions. As pedagogical modalities pivot to online platforms, there is an observable augmentation in the diversity and volume of devices interfacing with educational repositories, thus expanding the vectors accessible to malevolent cyber entities. Learners and educators interfacing with pedagogic content *via* domiciliary or public internet conduits may encounter compromised network integrity, heightening exposure to nefarious activities such as interception attacks, clandestine surveillance, and illegitimate system access. Virtual learning infrastructures are repositories of copious amounts of sensitive data, encompassing the personal particulars of students, scholarly records, and secure access credentials. The proliferation of digital correspondence has been accompanied by an escalation in deceptive stratagems, such as phishing and complex social engineering tactics designed to exfiltrate sensitive information or deploy malicious software. Adopting personal apparatuses under the "Bring Your Own Device" (BYOD) introduces complexities in orchestrating and safeguarding these devices from a plethora of security perils, including but not restricted to malware and unmediated system flaws. A potential deficit in cybersecurity awareness exists among the scholastic populace, rendering them more prone to digital threats. A regiment of ongoing enlightenment and training is necessitated. Academic institutions employ an array of software ecosystems to facilitate remote learning, with inherent vulnerabilities that could be exploited to secure unauthorized ingress or disrupt educational operations. Notably, smaller educational institutions might be challenged by a paucity of resources or specialized knowledge to counteract these cybersecurity adversities sufficiently. In response to these exigencies, it is imperative for educational establishments to enact comprehensive cybersecurity frameworks encompassing the fortification of network defenses, the institutionalization of periodic security evaluations, extensive training for the academic community, utilization of secure and authenticated virtual learning interfaces, and stringent adherence to data privacy statutes.

The absence of established cybersecurity standards and governance poses a noteworthy challenge for higher education institutions, particularly within the context of Western Balkan countries. Higher education institutions are increasingly facing ransomware attacks, with a report indicating that nearly two-thirds (64%) of institutions experienced such attacks last year. The impact can be substantial, leading to operational disruptions

and financial losses. To defend against these, a Zero Trust security model is recommended, which emphasizes the need for explicit verification, least privileged access, and the assumption that a breach has already occurred or will soon occur (*Scholz, Hagen & Lee, 2021*). Many higher education institutions are in the early stages of their IAM journey, struggling with piecemeal approaches and the need to aggregate solutions from multiple vendors or address gaps from a single IAM vendor. The complexity of managing a large number of identities, especially in the context of remote learning, adds to these challenges (*Bio-Key International, 2022*). The pandemic has exacerbated budget cuts in the education sector, limiting the funds available for cybersecurity investments. This financial strain, combined with the challenge of protecting expansive and open college networks, makes institutions vulnerable to cyberattacks (*Miller, 2022*).

In accordance with the findings reported by *Stojanovic et al. (2021)* and *Gecevska, Lombardi & ČUŠ (2009)*, Western Balkan governments, while ostensibly demonstrating preparedness at a conceptual level, manifest a practical effectiveness deficit in handling cybersecurity attacks. The incidence of data breaches is on the rise across organizations within Western Balkan countries, with a pronounced emphasis on HEIs. The advent of the COVID-19 pandemic has further facilitated threat actors in infiltrating higher education networks, given the widespread provision of remote access to students and staff. *Chapman (2021)* highlight that a majority of Western Balkan countries have either established or lack an e-learning system.

Moreover, there is an identified inadequacy in cybersecurity expertise and incident response capabilities within HEI administrations. Observations indicate a notable vulnerability in a substantial proportion of eLMSs, characterized by the absence of robust cybersecurity processes for conducting vulnerability assessments. Additionally, the various technologies integrated into LMSs across Western Balkan countries lack a security-oriented design, and the absence of integrated standards exacerbates the overall susceptibility to cyber attacks. The imperative nature of risk management within HEIs is underscored by *Chapman (2021)*, who delineates essential inquiries for the evaluation of cybersecurity risk:

1. Are systems adequately patched and maintained with up-to-date security measures?
2. Is there a systematic implementation of routine vulnerability scans as part of a comprehensive vulnerability management policy?
3. Is there a well-defined incident response plan in place to address potential security breaches?
4. Do the monitoring and mitigation systems encompass relevant cybersecurity risks effectively?
5. Is the network provider proficiently mitigating denial-of-service attacks in alignment with cybersecurity objectives?

As a result, it is imperative to develop a specialized incident response strategy tailored to individual instances of eLMS in Western Balkan countries, as underscored by *Ramani (2017)* and *Invicti (2021)*. The deficiency of proactive cybersecurity controls within HEIs in the Western Balkan countries poses a direct threat to information security. Moreover, the scarcity of proficient personnel and dedicated security operation centers (SOCs) manifests as a pervasive challenge in the context of Western Balkan countries.

There is a focus on enhancing cybersecurity governance and capabilities, as evidenced by a rapid response project initiated by the European Union (EU), which includes Albania, Montenegro, and North Macedonia. This project involves strengthening cybersecurity governance, adjusting cybersecurity legislation, and enhancing the training of Computer Security Incident Response Teams (CSIRTs). The effort is not only aimed at improving the cybersecurity stance of these countries but also at achieving EU and international standards to foster better opportunities and regional exchanges on digital and cybersecurity cooperation. Higher education institutions in western Balkan were increasingly encountering cyber threats, and attacks are becoming more sophisticated, tailoring malicious content to local languages and contexts. Cybercrime remains the main threat, particularly malware, phishing, ransomware, and Distributed Denial of Service (DDoS) attacks. Furthermore, these efforts reflect a growing awareness and commitment to cybersecurity in the Western Balkans' higher education sector, acknowledging the importance of resilience in the face of evolving digital threats (*Jin & Klopfer, 2021*; *ISAC, 2022*; *Plantera, 2023*; *Maigre, 2022*).

## Motivations and contributions

This research endeavors to present a lightweight cybersecurity framework, encompassing proactive controls, with the aim of fortifying the security of eLMS within HEIs situated in Western Balkan countries. The primary aim is to employ a penetration test approach for the systematic identification of vulnerabilities within eLMS platforms, subsequently devising proactive controls tailored to address each identified vulnerability. This study addresses the following research questions:

- What vulnerabilities and weaknesses characterize the cybersecurity landscape of eLMSs in HEIs in Western Balkan countries?
- What methodologies can be employed to implement proactive measures based on the identified vulnerabilities?

In addressing these inquiries, a penetration testing methodology is adopted, leveraging the MITRE Attack framework and the Open Web Application Security Project (OWASP) methodology. Additionally, the primary contribution of this study lies in proposing a model for a lightweight framework designed to assist HEIs in prioritizing and identifying vulnerabilities, subsequently facilitating the implementation of proactive controls identified during the penetration testing process. Importantly, this lightweight framework is intended to provide support to inexperienced and understaffed HEIs, thereby enhancing their cybersecurity infrastructure. A penetration testing methodology is employed to tackle these issues, utilizing the MITRE Attack framework and the OWASP methodology. Furthermore, the primary contribution of this study is to propose a lightweight framework model to assist HEIs in prioritizing and discovering vulnerabilities and implementing proactive controls found during the penetration test process. Furthermore, the lightweight framework will support inexperienced, understaffed HEIs and improve their cybersecurity infrastructure.

### Article organization

The rest of the article is structured as follows. 'Methodology' describes the proposed methodology for penetration tests usable to find the vulnerabilities in eLMSs of HEIs. 'Proposed Lightweight Framework with Proactive Controls for ELMS' discusses the details of implementing the proposed lightweight framework for cybersecurity attack management. 'Results and Discussions' discusses the simulation of the proposed framework. 'Conclusion' concludes the article with some valuable remarks and suggestions.

## METHODOLOGY

A penetration test is one of the most common methodologies for determining vulnerabilities in different platforms. According to *Alghamdi (2021)*, penetration testing is crucial for identifying security vulnerabilities, but it is becoming sophisticated and time-consuming, resulting in poor reporting. An HEI can use various penetration test principles and procedures for such purposes. Penetration testing methodologies, such as PTES, are used by organizations to identify and evaluate the security of their systems, networks, and applications. Some use cases for penetration testing are given below.

- **Risk management:** Penetration testing can identify potential vulnerabilities and risks that attackers could exploit, allowing organizations to prioritize and address the most critical issues.
- **Auditing:** Organizations may conduct regular penetration testing as part of their internal audit processes to ensure that their security controls are working as intended.
- **Incident response:** Penetration testing can be used to simulate real-world attacks and test an organization's incident response procedures to identify any gaps and make improvements (*Alexei, Nistiriuc & Alexei, 2020*).
- **Insider threat identification:** Penetration testing can help to find potential insider attacks by testing user access controls and evaluating the impact of compromised credentials.

Penetration testing methodologies, such as PTES, are used by organizations to identify and evaluate the security of their systems, networks, and applications. Some use cases for penetration testing are given below.

- **Risk management:** Penetration testing can identify potential vulnerabilities and risks that attackers could exploit, allowing organizations to prioritize and address the most critical issues.
- **Auditing:** Organizations may conduct regular penetration testing as part of their internal audit processes to ensure that their security controls are working as intended.
- **Incident response:** Penetration testing can be used to simulate real-world attacks and test an organization's incident response procedures to identify any gaps and make improvements (*Alexei, Nistiriuc & Alexei, 2020*).
- **Insider threat identification:** Penetration testing can help to find potential insider attacks by testing user access controls and evaluating the impact of compromised credentials.

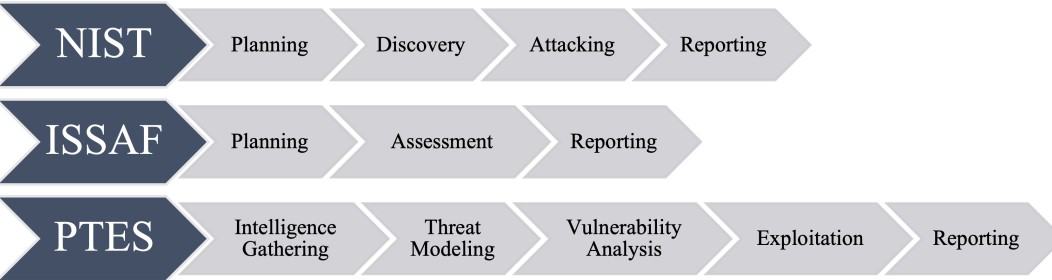

**Figure 1** Penetration test phases NIST, ISAAF, and PTES.

- **Internal network testing:** OSSTMM provides a comprehensive methodology for testing internal networks, including guidelines for information gathering, vulnerability scanning, vulnerability assessment, and penetration testing.

The Open-Source Security Testing Methodology Manual (OSSTMM) is a systematic and structured security testing methodology used in various contexts to evaluate the security of systems, networks, and applications. Some use cases for OSSTMM include:

- **Web application testing:** OSSTMM provides a methodology for testing web applications, including guidelines for identifying and exploiting vulnerabilities in web applications.
- **Third-party testing:** Organizations may use OSSTMM to conduct security testing of third-party vendors, such as service providers, to ensure that they meet the organization's security standards (*Sekulovic, 2018*).

Open Information Systems Security Group (OISSG) is a community-driven organization that provides guidelines, methodologies, and best practices for security testing. The OISSG's guidelines can be used in a variety of contexts. Some of the use cases for OISSG include:

- **Penetration testing:** OISSG provides guidelines for conducting penetration testing and identifying vulnerabilities attackers could exploit.
- **Risk management:** OISSG provides guidelines for identifying and evaluating the risks associated with different systems, networks, and applications and developing strategies for mitigating those risks.
- **Incident response:** OISSG provides guidelines for incident response procedures and identifying vulnerabilities that attackers could exploit (*Abu-Dabaseh & Alshammari, 2018*).

In addition, penetration test as a security mechanism are limited to only finding vulnerabilities, not fixing them or proposing specific preventive strategies (*Doyle et al., 2020*; *Korniyenko et al., 2021*). An overview of penetration test phases from various standards such as NIST, ISAAF, and PTES is shown in Fig. 1.

The comparison of different cybersecurity frameworks or methodologies, specifically NIST, ISSAF, and PTES:

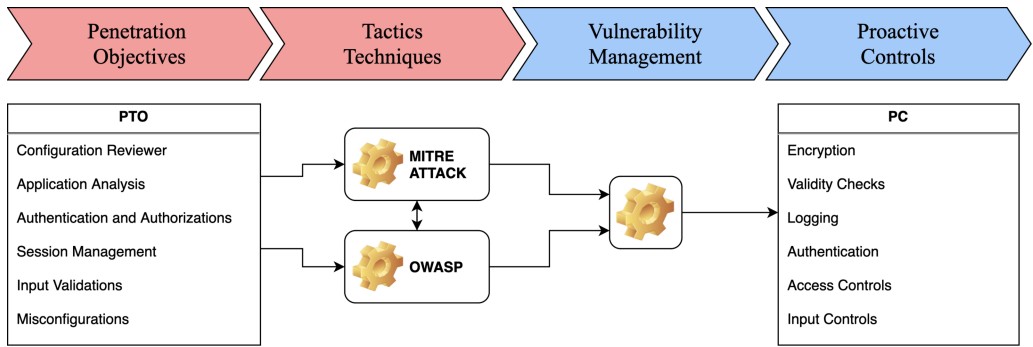

**Figure 2**  **Proposed lightweight framework with proactive controls for LMS.**

- **National Institute of Standards and Technology (NIST)** follows a series of steps, starting with Planning, moving to discovery, then Attacking, and finally, Reporting. This approach is very structured and is known for its comprehensive nature, focusing on protecting and maintaining the security of information systems.
- **Information Systems Security Assessment Framework (ISSAF)** is simplified into three stages: Planning, Assessment, and Reporting. It provides a streamlined approach to evaluating information system security, which can benefit organizations with limited cybersecurity resources.
- **Penetration Testing Execution Standard (PTES)** outlines a more detailed approach, starting with Intelligence Gathering, moving to Threat Modeling, then Vulnerability Analysis, followed by exploitation, and concluding with Reporting.

This article suggests a lightweight framework to protect higher education assets from various cybersecurity attacks. A lightweight framework can be a practical methodology for developing proactive controls based on a penetration testing approach to effectively protect the eLMS from various cybersecurity attacks, as illustrated in Fig. 2. The framework is divided into two parts. The first part pertains to the offensive aspect and encompasses penetration test objectives and tactics based on the MITRE ATT&CK framework and OWASP methodologies. The second part focuses on vulnerability management, including filtering and implementing proactive controls based on the vulnerabilities identified.

This framework can be implemented in several ways, including:

- **Vulnerability identification:** The framework can be used to identify vulnerabilities in systems, networks, and applications through penetration testing. This can consist of identifying known vulnerabilities, such as those listed in the Common Vulnerabilities and Exposures (CVE) database, and discovering new vulnerabilities.
- **Risk prioritization:** The framework can be used to prioritize the identified vulnerabilities based on their likelihood and potential impact. This allows organizations to focus on addressing the most critical vulnerabilities first.
- **Proactive controls:** The framework can be used to develop and implement proactive controls to mitigate the identified vulnerabilities. This can include implementing

security controls such as firewalls, intrusion detection and prevention systems, and patch management.

- **Continuous monitoring:** The framework can be used to continuously monitor systems, networks, and applications for vulnerabilities and threats, allowing organizations to identify and respond to new or emerging threats quickly.
- **Reports and metrics:** The framework can be used to generate reports and metrics that provide visibility into the organization's security posture, allowing them to track the effectiveness of their security controls over time.

The diagram suggests a comprehensive strategy that aligns cybersecurity objectives with the institution's infrastructure, utilizing well-known cybersecurity frameworks to structure the institution's approach to managing cyber risks. The framework emphasizes a balance between proactive measures to prevent incidents and reactive measures to respond to them, with an underlying acknowledgment of the need for resource allocation, including personnel and budget considerations. The framework is structured in three layers, suggesting a hierarchical approach to cybersecurity.

### Layer 1: strategic overview
- Penetration test objectives establishing the goals of penetration testing to identify security weaknesses.
- Tactics, Techniques, Tools, and Procedures (TTTP) define the methods and tools that will be used in penetration testing and incident response.
- Vulnerability management consists of identifying, classifying, prioritizing, remediating, and mitigating software vulnerabilities.
- Proactive controls prevent security incidents before they occur, such as patch management, strong authentication, and user education.

### Layer 2: operational details
- Reconnaissance testing scope determines the scope of the security testing, likely focused on gathering information about potential targets and vulnerabilities.
- MITRE ATT & CK and OWASP established cybersecurity frameworks (MITRE ATT & CK) and guidelines (OWASP) for known tactics, techniques, and procedures attackers use.
- Assessment analysis prioritizes the results of the security assessments to focus on the most critical vulnerabilities or areas of improvement.
- Control type, implementation actor, and costs consist of the control measures to be implemented, who will be responsible for implementation, and the associated costs.

### Layer 3: implementation and approach
- HEIs objectives guide the choice of security measures and priorities.
- HEIs infrastructure needs to be secured according to the framework.
- Framework approaches (Offensive and Defensive) incorporate both offensive measures (*e.g.*, ethical hacking, penetration testing) to identify vulnerabilities and defensive measures (*e.g.*, firewalls, intrusion detection systems) to protect against threats.

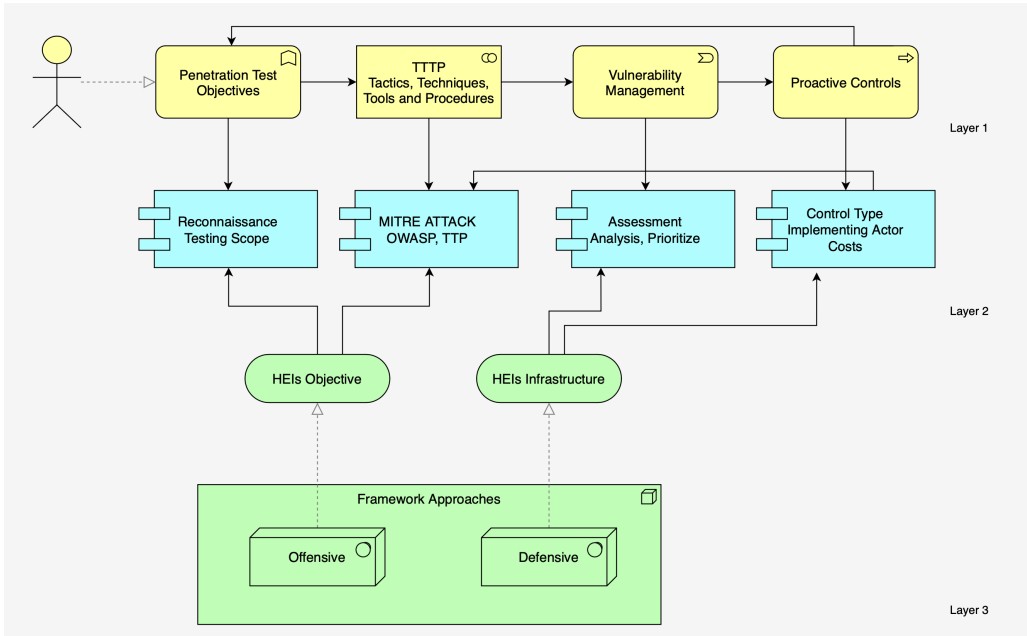

**Figure 3**  **The structure of the proposed framework.**

Overall, a lightweight framework can be a valuable tool for HEIs to identify and mitigate cybersecurity risks and proactively protect their assets from various cyberattacks, as shown in Fig. 2.

# PROPOSED LIGHTWEIGHT FRAMEWORK WITH PROACTIVE CONTROLS FOR ELMS

The methodology of the proposed framework encompasses four distinct implementation levels.Illustrated in Layer 1 (See Figs. 3, 4 and 5) is a comprehensive depiction of the lightweight framework, delineated by its constituent elements: penetration test objectives (PTO), penetration test processes (PTP), vulnerability management (VM), and proactive controls (PC).

 The proposed diagram suggests a strategic approach to cybersecurity, starting with penetration testing to identify vulnerabilities and using established frameworks to structure the approach to threat mitigation.

It emphasizes the importance of managing vulnerabilities and implementing proactive controls to improve overall security posture. The flow from penetration objectives to tactics and techniques, then on to vulnerability management and proactive controls, indicates a comprehensive process from identifying vulnerabilities to implementing measures to address them. The conceptual diagram is organized into four main components: PTO, tactics technique, vulnerability management, and PC.

- PTO list elements that are likely objectives or targets for penetration testing within an organization. These areas within an IT system or application would be scrutinized

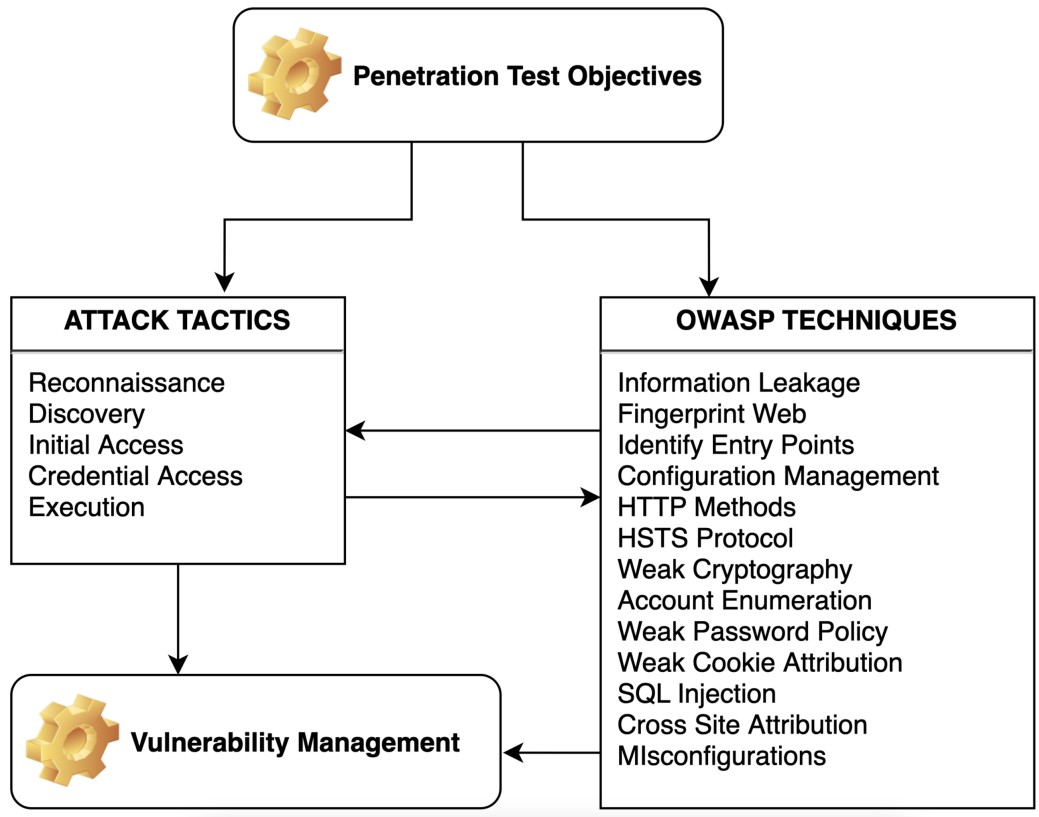

**Figure 4  The offensive approach of lightweight framework.**

during a penetration test to identify security weaknesses. It includes configuration review, application analysis, authentication and authorization, session management, input validations, and misconfigurations.

- Tactics technique refers to the specific tactics and methods employed during testing or as part of the security strategy. These frameworks provide structured approaches to identifying potential threats and the means to mitigate them. It might link to two well-known cybersecurity frameworks: MITRE ATT&CK and the Open Web Application Security Project (OWASP).
- Vulnerability management involves identifying, evaluating, prioritizing, and remedying software vulnerabilities to prevent exploitation.
- Proactive controls (PC) lists proactive measures that can be implemented to prevent security breaches, such as, encryption, validity checks, logging, authentication, access control, and input controls.

The diagram shows a feedback loop between the ATT & CK tactics and OWASP techniques and vulnerability management, suggesting that the outcomes of using these tactics and techniques feed into the process of managing vulnerabilities. This reflects a proactive approach to cybersecurity, where continuous penetration testing and assessment inform the ongoing process of managing and mitigating risks. The diagram represents

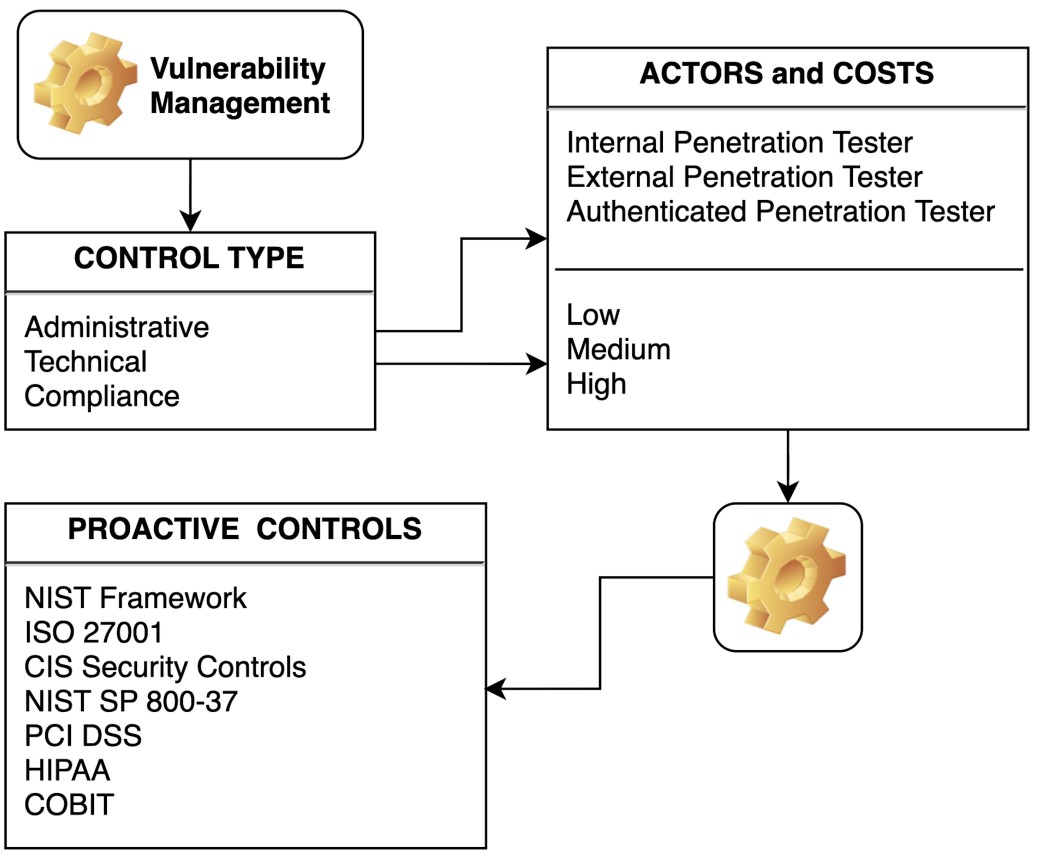

**Figure 5** The defensive approach of the proposed framework.

a cybersecurity approach focusing on penetration testing objectives, tactics, techniques, and vulnerability management. Penetration test objectives typically identify and exploit vulnerabilities to understand the security posture of a system. ATT & CK Tactics listed tactics—reconnaissance, discovery, initial access, credential access, and execution—stages an attacker might go through to compromise a system. These tactics help to structure penetration testing efforts by simulating real-world attack scenarios. OWASP techniques list common web application security issues identified by the Open Web Application Security Project (OWASP). These include problems like information leakage, weak cryptography, and common vulnerabilities such as SQL injection and Cross-Site Scripting (XSS). Penetration testers use these techniques to try and exploit vulnerabilities in web applications. Vulnerability Management is the process that follows the identification of vulnerabilities. It includes the prioritization, remediation, and mitigation of discovered vulnerabilities to enhance the security of the system.

The flowchart demonstrates how vulnerability management is informed by understanding the types of controls that can be applied, the actors involved, and the associated costs. It also shows that the process is underpinned by a range of industry frameworks and standards that guide the establishment of proactive controls

to prevent security breaches. Vulnerability management is the central element of the diagram, indicating that it is the core process being described. Vulnerability management encompasses identifying, classifying, prioritizing, remedying, and mitigating vulnerabilities. Control type breaks down the types of controls that can be implemented as part of vulnerability management into three categories:

- **Administrative:** Policies, procedures, and other managerial controls.
- **Technical:** Hardware or software mechanisms that enforce or monitor security.
- **Compliance:** Controls to ensure adherence to laws, regulations, and policies.

Actors and costs suggest that implementing the controls from the "Control Type" box involves different actors, such as internal or external penetration testers and authenticated testers (who test the system with valid access credentials). Costs are considered on a scale of low, medium, to high, indicating that the choice of actor and the type of control implemented will affect the overall cost of the vulnerability management process. Proactive controls lists various established cybersecurity frameworks and standards organizations can use to inform their proactive control measures. These include NIST Framework, CIS Critical Security Control, ISO 27001 Information Security Management, NIST SP 800-37, NIST SP 800-37, PCI DSS, HIPAA rules, and COBIT.

A lightweight cybersecurity framework can be effectively applied globally, providing HEIs worldwide with the guidance needed to protect their information assets and the privacy of their students and staff in a cost-effective and pragmatic manner. The framework is designed to scale with the size and complexity of different institutions, from small colleges to large universities with multiple campuses, accommodate various types of educational institutions with different IT infrastructures, resources, and regulatory environments, and maximize cybersecurity benefits while minimizing costs and resource use, which is crucial for institutions with limited budgets. The framework is designed to help institutions comply with a range of international and local regulations, leverage internationally recognized standards and best practices, such as those from ISO, NIST, and OWASP, which are globally applicable and respected, and allow for regional customization to address specific threat landscapes effectively.

In subsequent iterations, this methodology endeavors to automate and integrate a broader spectrum of vulnerabilities and proactive controls through an in-depth analysis of eLMSs situated in Western Balkan countries. This risk assessment process is tailored to be lightweight by focusing on the most significant threats and vulnerabilities that could affect eLMS platforms rather than a comprehensive assessment of all possible risks. The goal is to maintain a balance between thorough risk management and the practical constraints of the HEIs in the Western Balkans. Risk assessment can be seen as a critical first step in the defense mechanism in the context of the proposed lightweight framework model for protecting eLMS in HEIs. The assessment would typically involve identifying, analyzing, and evaluating the potential risks that eLMS platforms might face. The model includes a structured approach to risk assessment. Determine which assets are critical to the eLMS platform's operations. This includes software, data, hardware, and services essential for the eLMS to function. Identify potential cybersecurity threats that could affect the eLMS, such

as malware, phishing, DDoS attacks, or insider threats. This can be informed by threat intelligence and historical data. Use tools and techniques, possibly including those from the framework like open-source vulnerability scanners or checklists, to identify existing vulnerabilities within the eLMS, like outdated software or weak configurations. Evaluate the potential impact of each identified threat exploiting a vulnerability. This includes considering the consequences of data breaches, service interruptions, and compliance violations. Determine the probability of each threat materializing by considering factors such as the HEI's location, the sophistication of potential attackers, and the current geopolitical climate. Combine the impact and likelihood to rate the level of risk each threat poses to the eLMS platform. Risks are often categorized as low, medium, or high. Review current cybersecurity controls to assess their effectiveness against identified risks. Determine if additional measures are needed and what those might be. Based on the evaluation, prioritize the risks that require immediate attention and those that can be monitored over time. Resource allocation should focus on high-priority risks. Cybersecurity is dynamic, so the risk assessment process should be iterative. Regularly review and update the risk assessment to account for new threats, vulnerabilities, and changes in the HEI's environment. The proposed lightweight framework fortifies eLMS against cybersecurity threats in HEIs, particularly within the context of the Western Balkans (*Maigre, 2022*; *Henry, 2020*; *Castelo, 2020*).

Managing access control and identity management through appropriate cryptographic protocols and schemes ensures the security of the eLMS platform. Using firewalls and network monitoring is a necessary part of a defensive strategy to protect against external threats. Resource optimization is a strategic imperative to ensure that limited financial and computational resources are used effectively and efficiently, providing the best possible security posture with the available assets. Access control and identity management are pivotal in ensuring that only authorized individuals can access educational institutions' systems and data. Utilizing strong authentication protocols like OAuth 2.0 for authorizing and authenticating users who are accessing the systems. Implementing MFA to add an extra layer of security. Define user roles and assign access rights accordingly. Use Transport Layer Security (TLS) and its predecessor, Secure Sockets Layer (SSL), for secure communication over a computer network. Ensuring that only necessary hardware and software are purchased and maintained, avoiding unnecessary expenditures. Leveraging cloud services can reduce costs related to on-premises data centers, such as power, cooling, and maintenance.

## RESULTS AND DISCUSSIONS

The eLMSs within the Western Balkan countries govern a diverse array of student activities, encompassing assignments, file transfers, faculty interactions, and subject-related engagements. These systems comprise stored data, including student and lecturer profile pages, functionalities for the management of files through downloading and uploading various documents, and virtual learning-relevant forum pages. The present research employs a qualitative and descriptive methodology (*Invicti, 2021*) based on penetration

Table 1  Vulnerabilities discovered in Western Balkan HEIs eLMS.

| Common Vulnerabilities | T1 | T2 | T3 | T4 | T5 | T6 | T7 | T8 |
|---|---|---|---|---|---|---|---|---|
| Fingerprint web-server | × | | × | | × | × | | × |
| Trace method enabled | | × | | | | | | × |
| SSL support v2 | × | | × | | × | × | | × |
| SSL weak cipher | | × | | | | × | | × |
| SSL expired certificate | | | × | | × | × | | |
| SSL self-signed certificate | | | × | | | | × | |
| Credentials over GET methods | × | | | | × | | | × |
| Lack of client-side validations | × | | | | × | | | × |
| Lack of server-Side validations | | | × | | | | × | |
| Weak lockout mechanism | | × | | | | × | × | |
| Session ID prediction | | × | × | × | | × | | |
| Weak password Policy | | | | | | | | × |
| Weak cookie settings | × | | × | | | × | | × |
| Email Enumerations through Forms | × | | | × | | | | |
| XSS in URL parameters | | | | × | × | | | × |
| Lack of "http-only" and "security" flags | | × | | | | | | × |
| Cookie expires | | × | | | × | × | | |
| SQL Injections | | | | | × | | × | × |
| Unnecessary features | | × | | × | | × | | × |
| Unpatched software | × | | × | | × | | | |

testing techniques (*Al-Shaer, Spring & Christou, 2020*; *Korniyenko et al., 2021*; *The MITRE Corporation, 2021*; *OWASP, 2021b*; *Yosifova, 2021*) to identify essential vulnerabilities in eLMS in Western Balkan HEIs. The primary objective is to discern critical vulnerabilities within eLMSs in HEIs situated in the Western Balkan region. The study focuses on eight HEIs spanning four Western Balkan countries: Albania, Kosovo, Montenegro, and North Macedonia. The process of penetration testing (*Zakaria et al., 2019*) involves passive analysis of eLMS for weaknesses, technical flaws, or vulnerabilities. According to MITRE ATT&CK and OWASP (*Al-Shaer, Spring & Christou, 2020*; *Korniyenko et al., 2021*; *Pham & Dang, 2018*; *Zare, Zare & Azadi, 2018*), the primary purpose of penetration tests is to find more effective attack vectors as well as exploit vulnerabilities.

The methodology outlined is utilized to obtain all of the results listed in Table 1, which pertain to eight targets from countries in the Western Balkans. Most eLMS targets are vulnerable due to untrustworthy inputs, weak cryptography, lack of client/server-side controls, weak authentication mechanisms, misconfigurations, and open unnecessary features.

Table 1 represents a summary of common vulnerabilities found across various systems, as identified through a series of tests (T1 through T8) conducted as part of a cybersecurity audit within the context of a lightweight framework for higher education institutions in the Western Balkan countries. Table 1 lists several types of vulnerabilities, such as issues with Secure Sockets Layer (SSL) configurations, problems with session management (like Session ID Prediction), and other common web application vulnerabilities like SQL

Injections and Cross-Site Scripting (XSS). Each "X" in Table 1 indicates the presence of the vulnerability in the corresponding test.

- Fingerprint web server refers to the ability to detect the type and version of a web server by sending it requests and analyzing the responses, which could give attackers information on potential vulnerabilities.
- The trace method enabled is used for diagnostic purposes and can be exploited by attackers to gain access to information in HTTP headers, such as cookies and authentication data.
- SSL support v2 indicates that the outdated and insecure version 2 of the SSL protocol, which has known vulnerabilities, is supported.
- SSL weak cipher refers to the use of encryption algorithms that are no longer considered strong and can be easily broken by attackers.
- SSL expired certificate use an expired SSL certificate can lead to man-in-the-middle attacks as the website's identity cannot be confirmed.
- SSL self-signed certificates are not trusted by default and can be a sign of a potential man-in-the-middle attack.
- Credentials over GET methods consist of passing sensitive information such as login credentials in the URL (*via* GET requests) that can expose them to anyone with access to the URL.
- Without client-side/server-side validation, an application may accept malicious input that can lead to various attacks.
- A weak lock-out mechanism does not lock out users after multiple failed login attempts, making it vulnerable to brute force attacks.
- Session ID Prediction makes it easier for attackers to hijack user sessions.
- A weak password policy may choose passwords that are easy to guess or brute force.
- Weak cookie settings can allow attackers to intercept or manipulate cookies.
- Email enumerations through forms reveal whether an email address is associated with an account; it could aid an attacker in crafting a targeted attack.
- XSS in URL parameters refers to cross-site scripting vulnerabilities that occur when an application includes unvalidated or unescaped user input in URLs.
- Lack of "http-only" and "security" flags can be accessed by client-side scripts, which could lead to cross-site scripting attacks.
- SQL injections can insert or manipulate SQL queries through the application, potentially gaining access to or manipulating the database.
- Unnecessary features can introduce additional security risks
- Unpatched software has not been updated with the latest patches and may have known vulnerabilities that can be exploited.

In terms of the lightweight cybersecurity framework's application to the Western Balkan HEIs, this table shows that there are multiple common points of weakness that need to be addressed. The recurrence of "X" marks across the tests for each vulnerability indicates a pattern that may suggest systemic flaws in their e-learning platform web applications. Considering the potential lack of skilled cybersecurity professionals in the

region, a lightweight framework would prioritize addressing these vulnerabilities in a cost-effective and resource-efficient manner. The results suggest that Western Balkan HEIs could significantly improve their cybersecurity resilience by focusing on these common vulnerabilities and using a tailored lightweight framework that considers the region's specific challenges and constraints.

As asserted by *Alexei, Nistiriuc & Alexei (2020)*, within a survey encompassing thirty scholarly articles, five researchers within the academic domain advocate for the integration of the ISO27001 standard in HEIs, while two recommend the adoption of the COBIT framework. In addition, two articles suggest the utilization of the COBIT framework, one advocates for the incorporation of ITIL best practices, and another proposes a hybrid approach. Conversely, a significant proportion of researchers present their individualized strategies for enhancing cybersecurity within HEIs. Moreover, the envisaged methodology tailored for HEIs in the Western Balkans seeks to identify vulnerabilities within eLMSs and proactively establish controls. This approach integrates the MITRE ATT&CK framework, the OWASP methodology, CIS controls, and the NIST framework to formulate a comprehensive set of proactive controls. Based on a real-world scenario, we will propose many approaches to avoid failed login attempts to an eLMS. Implement account lockout to limit the number of consecutive failed login attempts and lock the account for a certain period. Implement a CAPTCHA system to prevent automated attempts to login. Implement multi-factor authentication by requiring multiple forms of authentication. Block I.P. addresses that have a high number of failed login attempts. Implement a security information and event management (SIEM) system to monitor the system for unusual login patterns and alert the administrator if there are too many failed login attempts. Asking security questions, such as personal information, after a certain number of unsuccessful attempts can help verify the user's identity. Implementing two-factor authentication (2FA) in a learning management system can help to improve security by adding an extra layer of protection. Keep track of all login attempts, both successful and unsuccessful, and regularly review the logs to detect any suspicious activity. The framework is designed to be flexible and salable to adapt to the changing threat landscape and the evolving needs of educational institutions. It aims to provide a balanced approach that ensures the security of assets without imposing excessive administrative or financial burdens on the institutions. Under the proposed lightweight cybersecurity framework for HEIs in the Western Balkans, the protection of assets would be approached through several strategic layers: asset identification and classification, access control, data encryption, regular security audits and vulnerability assessments, patch management, security awareness training, incident response plan, backup and recovery, network security, and compliance with legal and regulatory requirements.

## CONCLUSION

Implementing e-learning management systems (eLMS) without due consideration for security can lead to many challenges, especially in regions like the Western Balkans, where resources may be limited and cybersecurity awareness may not be widespread. Addressing

these challenges requires a comprehensive approach involving policy development, investment in technology, training and awareness programs, and collaboration between educational institutions, government bodies, and cybersecurity experts. E-learning systems store significant personal data from students and staff, such as names, addresses, academic records, and sometimes even payment information. Inadequate security measures can lead to data breaches, risking the exposure of sensitive information. An eLMS without robust security features is an attractive target for cyberattacks, ranging from denial-of-service attacks disrupting access to learning materials to more severe ransomware attacks that encrypt valuable data. Without a proper security framework, an institution may not have an incident response plan in place. This can severely hamper its ability to respond to and recover from cyber incidents quickly. E-learning platforms host proprietary course materials and research data. Without adequate security, there is a risk of unauthorized access and intellectual property theft. The eLMS must continuously update and maintain to address new security threats. Neglecting this can leave systems vulnerable to new types of cyberattacks. There is a lack of awareness or a cultural barrier to understanding the importance of cybersecurity. This can result in a casual approach to security among students and staff, exacerbating vulnerabilities. In the Western Balkans, financial and human resources might be scarce to invest in advanced cybersecurity infrastructure and to train staff adequately on cybersecurity best practices. In the context of academic research, the proposal of a lightweight framework model to protect eLMS in HEIs in the Western Balkans is predicated on several core academic concepts and methodologies. The proposed lightweight framework designed for the Western Balkans higher education institutions, compared to NIST, ISSAF, and PTES, highlights the advantages:

- The lightweight framework might be specifically tailored to the common threats and resources available in the Western Balkan higher education environment, providing a more focused and relevant approach than the general methodologies.
- Given the resource constraints in the region, a lightweight framework would likely simplify the process to focus on the most impactful activities, reducing the complexity and cost associated with more comprehensive standards.
- It could emphasize essential controls that offer the most significant security benefit, particularly valuable in an environment where institutions may not be able to implement a broad array of measures.
- The framework could incorporate education and cybersecurity awareness elements, which are crucial in environments where cybersecurity knowledge may not be widespread.
- The lightweight framework is likely designed for rapid deployment and agility, allowing institutions to improve their cybersecurity posture in response to emerging threats quickly.
- By focusing on the most significant risks and implementing key controls efficiently, a lightweight cybersecurity framework can provide a practical and cost-effective solution for improving cybersecurity resilience in resource-constrained environments like those of higher education institutions in the Western Balkans.

In conclusion, the article proposes a novel, lightweight cybersecurity framework designed to safeguard eLMS platforms in Western Balkan HEIs. This framework is grounded in an open-source methodology, allowing customization, community support, and cost-effectiveness. By integrating the framework as a self-assessment tool, HEIs can actively gauge and enhance their cybersecurity maturity. The iterative penetration testing process, a key component of the framework, is informed by empirical findings that underscore the prevalent vulnerabilities within eLMS platforms. The proposed model facilitates the implementation of proactive controls, emphasizing preventive measures over-reactive responses, aligning with best cybersecurity management practices.

### Funding
No funding was used in this study.

### Competing Interests
The authors declare there are no competing interests.

### Author Contributions
- Krenar Kepuska conceived and designed the experiments, performed the experiments, analyzed the data, performed the computation work, prepared figures and/or tables, authored or reviewed drafts of the article, and approved the final draft.
- Milo Tomasevic conceived and designed the experiments, performed the experiments, analyzed the data, performed the computation work, prepared figures and/or tables, authored or reviewed drafts of the article, and approved the final draft.

### Data Availability
   The data used to support the findings of this study are included in the article. The code is available in the Supplementary File.

### Supplemental Information
Supplemental information for this article can be found online at http://dx.doi.org/10.7717/peerj-cs.1958#supplemental-information.

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
