# Peer review of "A lightweight framework for cyber risk management in Western Balkan higher education institutions"

_PeerJ Computer Science, doi:10.7717/peerj-cs.1958_

## Round 0.1 · original submission · Major Revisions

The review process is now complete. While finding your paper interesting and worthy of publication, the referees and I feel that more work could be done before the paper is published. My decision is therefore to provisionally accept your paper subject to major revisions.

**Language Note:** PeerJ staff have identified that the English language needs to be improved. When you prepare your next revision, please either (i) have a colleague who is proficient in English and familiar with the subject matter review your manuscript, or (ii) contact a professional editing service to review your manuscript. PeerJ can provide language editing services - you can contact us at [email protected] for pricing (be sure to provide your manuscript number and title). – PeerJ Staff

Reviewer 1 ·

Basic reporting

Contextualization and Motivation:
The introduction should provide a more detailed contextualization of the prevalence of data breaches in Higher Education Institutions (HEIs) globally, with specific examples or statistics to underscore the urgency of the issue. This will enhance the readers' understanding of the broader landscape and the significance of the proposed cybersecurity methodology.

Focus on Post-COVID Era:
While the paper acknowledges the shift to online learning platforms in the post-COVID era, it would be beneficial to elaborate on how this transition has exacerbated cybersecurity challenges in HEIs. Discuss the unique vulnerabilities introduced by the rapid adoption of online platforms to provide a more comprehensive understanding.



Local Contextualization for Western Balkans:
Offer a more nuanced explanation of the specific challenges faced by HEIs in the Western Balkans. Provide insights into the region's socio-political and technological landscape, emphasizing why incident response plans are notably absent and how this absence impacts cybersecurity resilience.


Comprehensive Framework Explanation:
Enhance the clarity and detail of the proposed cybersecurity methodology, termed the "lightweight framework with proactive controls." Provide a step-by-step breakdown of the framework components, how it identifies vulnerabilities, and the rationale behind the proactive controls suggested. This will assist readers in grasping the practical application of the methodology.


Broader Applicability:
Discuss the potential applicability of the proposed framework beyond the Western Balkans. Consider how the methodology's principles could be adapted and adopted in diverse global contexts, thereby contributing to the broader discourse on cybersecurity in HEIs.

Experimental design

In-depth Analysis of Cybersecurity Landscape:
Expand the section discussing recent studies on information security in HEIs. Provide a deeper analysis of the types of cyber attacks observed, the specific vulnerabilities targeted, and the implications for academic institutions. This will contribute to a more thorough foundation for the proposed framework.

Incorporate Real-world Examples:
Consider integrating real-world examples or case studies where the proposed framework has been successfully applied or scenarios where it could have prevented security incidents. This will help readers visualize the practical impact of the methodology.

Validity of the findings

Detailed Exploration of Existing Gaps:
Elaborate further on the challenges associated with the implementation of e-learning management systems without due consideration for security in the Western Balkans. Highlight specific instances or case studies to underscore the severity of the issue and provide a basis for the proposed framework's relevance.

Reviewer 2 ·

Basic reporting

The article was written in English and used clear, unambiguous, technically correct text. Besides, the language needs polishing.

Experimental design

The article fails to meet PeerJ's standarts in terms of its experimantal results. This manuscript involves results for some areas cyber risk problems. The current state of the technology is not discussed in detail. There are already a few studies on this topic. For such a scientific work to contribute to the body of scientific knowledge, especially through a journal with these quality standards, it must propose a method or methodology to solve a global or general problem.

Validity of the findings

The conclusions was properly stated and connected to the original question investigated. Although the results given have been realized, these results have not established a field that will support different future scientific studies based on this study.

·

Basic reporting

The authors presented a detailed study about cyber risk assessments and management in Western Balkan higher education institutions.

Experimental design

Results, discussion-Result, and analysis are presented in detail.

Validity of the findings

Conclusion should include details about the results obtained.

Additional comments

The organization of the paper is good. The presentation of the paper is acceptable. However, the article is not fair in its present form. It needs some improvements. Therefore, the paper can be conditionally accepted, provided the following concerns must be resolved.
(1) The authors must address how the risk is assessed. What policies and compliances are to be incorporated to address them?
(2) How can the authors manage access control and identity management? What are the related cryptographic protocols/schemes that are needed to address them?
(3) Why is firewall and network monitoring needed?
(4) Do you feel that resource optimization is essential? Justify your claim.
(5) How will the assets of an educational institute be protected under your proposed framework?

Reviewer 4 ·

Basic reporting

The paper encompasses a framework that identifies security vulnerabilities in e-learning systems in Western Balkan countries. It provides guidelines on how the security testing should be conducted.

- There are companies conducting penetration tests, actively engaged in such practices. Their differences from this penetration test should be examined and emphasized in the contribution section. The contribution to the literature is not clear or is inadequate.

- Does the automation of proactive control and continuous monitoring systems include autonomously operating structures, such as an IDS? This part is not clearly specified.

- Nessus, Metasploit, Nmap, SQLMap, Burp Suite, Wireshark. These applications are the fundamental tools used in standard penetration testing. The difference from a standard penetration test should be specified.

- The factors that make the proposed framework lightweight should be explained in detail.

- The conclusion section should be expanded.

- The references need to be reviewed.

Experimental design

- Experimental studies of the proposed framework should be conducted, and detailed results should be provided. Concrete data and comparisons demonstrating that the framework is lightweight should be presented.

Validity of the findings

- In the "Results and Discussion" section, the differences between the results of this framework and traditional penetration testing applied by companies or other similar frameworks should be evaluated. A comparison should be made, and based on this, advantages and disadvantages should be assessed.

---

## Round 0.2 · accepted · Accept

We are happy to inform you that your manuscript has been accepted for publication since the reviewers' comments have been addressed.

Reviewer 1 ·

Basic reporting

Authors updated the paper.

Experimental design

Authors updated the paper.

Validity of the findings

Authors updated the paper.

Reviewer 2 ·

Basic reporting

It is clear and has a fluent language feature. The background and the subject are provided in the sense of the area. The references and the cited studies are sufficient. Moreover, it has quality standarts for this type of journal.

Experimental design

It meets the standard of the journal of PeerJ in terms of experimental results and composition.

Validity of the findings

No comment.

Reviewer 4 ·

Basic reporting

The authors have made the necessary edits.

Experimental design

The authors have made the necessary edits.

Validity of the findings

The authors have made the necessary edits.